# rfaRm: An R client-side interface to facilitate the analysis of the Rfam database of RNA families

**Lara Sellés Vidal** [1]*, **Rafael Ayala**[2], **Guy-Bart Stan**[1]*, **Rodrigo Ledesma-Amaro**[1]*

**1** Department of Bioengineering, Faculty of Engineering, Imperial College London, London, United Kingdom,
**2** Department of Infectious Disease, Faculty of Medicine, Imperial College London, London, United Kingdom

* lara.selles12@imperial.ac.uk (LSV); g.stan@imperial.ac.uk (GBS); r.ledesma-amaro@imperial.ac.uk (RLA)

**Data Availability Statement:** The SARS-CoV-2 genome is available on the RefSeq database with accession number NC_045512.2. The sequence of the mitochondrial DNA of Ashbya gossypii is

## Abstract

rfaRm is an R package providing a client-side interface for the Rfam database of non-coding RNA and other structured RNA elements. The package facilitates the search of the Rfam database by keywords or sequences, as well as the retrieval of all available information about specific Rfam families, such as member sequences, multiple sequence alignments, secondary structures and covariance models. By providing such programmatic access to the Rfam database, rfaRm enables genomic workflows to incorporate information about non-coding RNA, whose potential cannot be fully exploited just through interactive access to the database. The features of rfaRm are demonstrated by using it to analyze the SARS-CoV-2 genome as an example case.

## Introduction

The Rfam database [1] is a collection of families of non-coding RNA and other structured RNA elements. Each family is defined by a multiple sequence alignment of a representative set of family members (seed alignment), a consensus secondary structure, and a covariance model [2], which integrates both multiple sequence alignment information and secondary structure information, and are analogous to the hidden Markov models used in Pfam [3] to determine the consensus sequence of protein families.

The database can be used to identify non-coding RNA elements within a nucleotide sequence of interest by searching it against the Rfam library of covariance models with the Infernal software [4]. Additionally, the database can be used to browse existing RNA families via keyword-based searches or direct access with the accession number or ID of specific families. Different pieces of information can be retrieved for each RNA family, including a descriptive summary, secondary structure information and consensus sequence, amongst many others. All of these functionalities can be accessed through the Rfam web-based interface. However, even though a RESTful API for the database is available, no client-side interface has been implemented so far to allow automated access to it, which makes any medium- or large-scale analysis a laborious and time-consuming process.

available on the RefSeq database with accession number NC_005789.1.

**Funding:** RLA, GBS and LSV acknowledge the ERA CoBioTech UKRI/BBSRC project SyCoLim (BB/T011408/1), Biotechnology and Biological Sciences Research Council (BBSRC) (https://bbsrc.ukri.org/). GBS acknowledges the EPSRC Fellowship for Growth (EP/M002187/1), Engineering and Phyisical Sciences Research Council (https://epsrc.ukri.org/). The funders had no role in study design, data collection and analysis, decision to publish, or preparation of the manuscript.

**Competing interests:** The authors have declared that no competing interests exist.

Here, we present a client-side interface to the Rfam database, enabling its programmatic access and therefore expanding the scope of the genomic analysis that can be carried out with the information provided by the Rfam database. The language of choice was R. This choice is based on the large number of tools already available in the Bioconductor project for the analysis of high-throughput genomic data. The software presented here complements these tools and facilitates the integration of the data retrieved via rfaRm within existing genomic workflows [5].

## Implementation

### Software features

rfaRm provides two types of functionalities: searches within the Rfam database, and retrieval of data associated to specific RNA families.

In its current version, rfaRm allows two types of searches within the Rfam database: by keyword, and by sequence. In a keyword search, the user can provide a keyword that will be matched against family descriptions and identifiers. Matching families are returned as a list of Rfam accession numbers. In a sequence search, the user submits an RNA sequence and the list of RNA families present in this sequence is returned. While the current implementation of the Rfam web server allows for queries on sequences of up to 10,000 nucleotides, rfaRm imposes no limit on the length of sequences to be analyzed. Instead, if a sequence longer than 10,000 nucleotides is provided as input, it is internally split into smaller, overlapping fragments that are then used to perform individual searches. Found hits are mapped back into the original sequence before being returned to the user.

Furthermore, rfaRm also provides a functionality analogous to the "clan competition" feature employed by the Rfam web server to ensure hit quality. If such functionality is enabled, groups (clans) of related Rfam families are defined. If two hits overlap by a user-defined length (by default, 50% of the length of the shortest hit), and they belong to the same clan, only the hit with best score is kept. rfaRm allows clan competition to be disabled, which in some cases might be preferrable to identify nested non-coding RNA hits.

After identifying a set of RNA families of interest, rfaRm allows to retrieve and plot different data about each family by providing their Rfam accession number or ID. The data that can be retrieved for each family include: a descriptive summary, the consensus sequence and secondary structure (in extended dot-bracket or WUSS notations) (Fig 1A), the covariance model, the seed alignment used to define it, several types of secondary structure plots (Fig 1B), the phylogenetic tree associated with the seed alignment (Fig 1C), the full list of sequence regions belonging to the family (including their GenBank accessions and starting and ending positions), and a list of entries of the PDB database with associated 3D structures for members of the family. Insightful plots of secondary structure and phylogenetic trees can be either directly displayed in R or saved into separate files with a desired format (including SVG to facilitate further edition).

### Integration with other packages and software

In order to facilitate the analysis and further manipulation of the data retrieved from Rfam, we designed rfaRm to output all data into standard formats that can be directly read by other software and R packages. All plots can be written as character strings representing SVG objects in XML format (which can be manipulated with the rsvg [6] and magick [7] R packages), in addition to being saved in many commonly used image formats. Consensus sequence and secondary structure in the extended dot-bracket format can be saved to files directly readable by the *R4RNA* R package [8] (Fig 2). Covariance models are outputted in the Infernal format. Seed

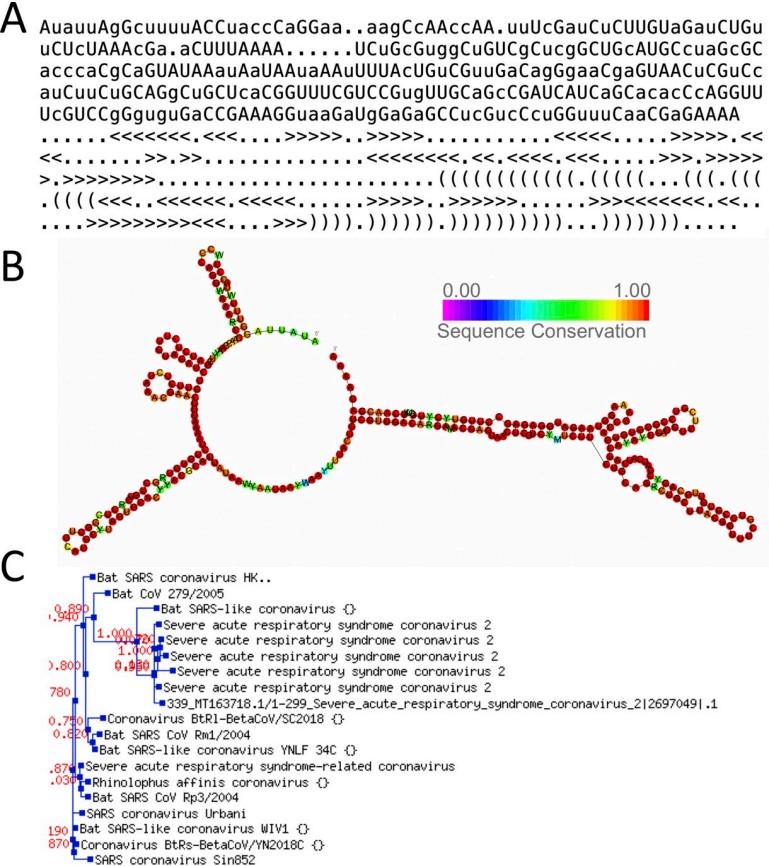

**Fig 1. Examples of data retrieved with rfaRm.** All data were retrieved for the Rfam family RF03120, comprising the SARS beta-coronavirus 5'-UTR. (A) Consensus sequence and secondary structure. (B) Secondary structure plot colored by sequence conservation. (C) Phylogenetic tree of the seed alignment used to define the family.

multiple sequence alignments can be stored as Biostrings Multiple Alignment R objects (one of the standard formats of the Bioconductor project) [9] or saved into FASTA- or Stockholm-formatted files. Finally, phylogenetic trees can be saved in the New Hampshire Extended format (NHX), which can be read by a variety of software such as the treeio R package [10].

## Description of available functions

A short description and an example of usage for each of the functions available in rfaRm is presented here. Further details can be found in the manual and vignette of the package. In the examples presented here, the mitochondrial DNA of *Ashbya gossypii* is used (RefSeq accession number NC_005789.1).

**rfamTextSearchFamilyAccession.** *Purpose.* Searches the Rfam database for entries containing a specified keyword in the family ID, summary or description.

*Arguments*

- *query*: string with the keyword to be searched in the Rfam database.

*Example of usage*
```
## Search Rfam families associated to the keyword "tRNA"
rfamTextSearchFamilyAccession(rfamFamily = "tRNA")
```

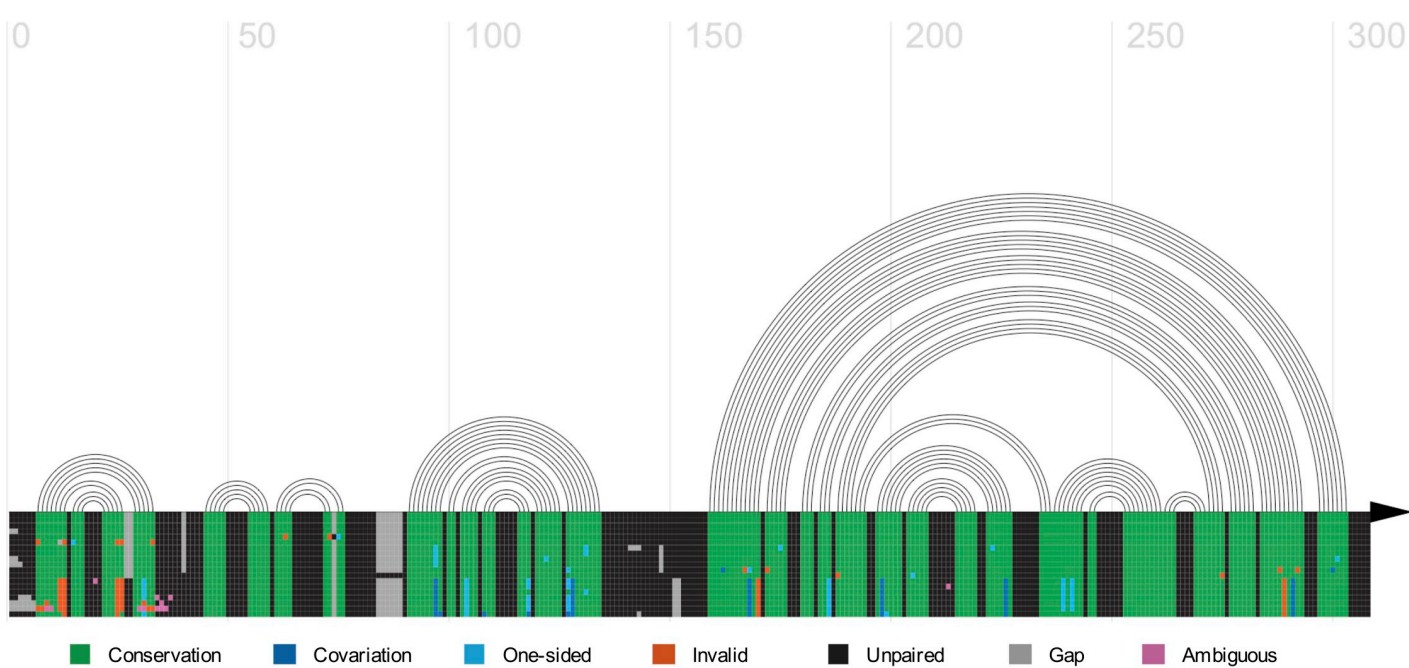

**Fig 2. Helix plot of the secondary structure of the 5' UTR of SARS beta-coronaviruses.** The helix plot was generated with the R4RNA package. Arcs indicate base pairings. There are 4 stem loops present (SL1, SL2, SL3 and SL4), as well as a larger structure known as SL5. The plot is annotated with information derived from the seed multiple sequence alignment. As expected, regions involved in base pairing are conserved.

**rfamSequenceSearch.** *Purpose*. Identifies non-coding RNA in a sequence provided by the user.

*Arguments*

- *sequence*: string with an RNA sequence to be searched against the Rfam database. Should contain only standard RNA symbols (i.e., "A", "U", "G" and "C").

- *fragmentsOverlap*: when a sequence larger than 10000 nucleotides is provided, it is internally split into smaller fragments before using them to search the Rfam database. This argument controls the number of overlapping bases between consecutive fragments.

- *clanCompetitionFilter*: logical value indicating if results should be reduced through a clan competition filter, which removes overlapping hits if they belong to Rfam families of the same clan and have an overlap above a certain threshold.

- *clanOverlapThreshold*: number indicating the minimum overlap between two hits (as a fraction of the smallest hit) to remove the hit with the worst e-value if their families belong to the same Rfam clan.

*Example of usage*
```
library("seqinr")
## Read the sequence of the mitochondrial DNA of Ashbya gossypii
fasta_a_gossypii_MT <- unlist(read.fasta("a_gossypii_MT.fasta", seq-
type = "DNA",
as.string = TRUE))
a_gossypii_MT_RNA <- gsub("t", "u", fasta_a_gossypii_MT, ignore.
case = TRUE)
## Search for non-coding RNA with clan competition filter enabled
a_gossypii_MT_hits_clanCompetition <- rfamSequenceSearch
(sequence = a_gossypii_MT_RNA,
```

```
fragmentsOverlap = 1000,
clanCompetitionFilter = TRUE)
## Count the number of detected non-coding RNA
length(a_gossypii_MT_hits_clanCompetition)
```

**rfamFamilyAccessionToID.** *Purpose*. Converts an Rfam family accession to the corresponding family ID.

*Arguments*

- *rfamFamilyAccession*: string with the Rfam family accession to be converted to a family ID.

  *Example of usage*
  ```
  ## Extract the Rfam family accession of the first hit
  ## detected in the mitochondrial DNA of Ashbya gossypii
  testAccession <- a_gossypii_MT_hits_clanCompetition[[1]]
  $rfamAccession
  ## Obtain the corresponding Rfam family ID
  rfamFamilyAccessionToID(rfamFamilyAccession = testAccession)
  ```

**rfamFamilyIDToAccession.** *Purpose*. Converts an Rfam family ID to the corresponding family accession.

*Arguments*.

- *rfamFamilyID*: string with the Rfam family ID to be converted to a family accession.

  *Example of usage*
  ```
  ## Extract the Rfam family accession of the first hit
  ## detected in the mitochondrial DNA of Ashbya gossypii
  testID <- a_gossypii_MT_hits_clanCompetition[[1]]$rfamID
  ## Obtain the corresponding Rfam family ID
  rfamFamilyAccessionToID(rfamFamilyID = testID)
  ```

**rfamFamilySummary.** *Purpose*. Retrieves a brief summary describing the specified Rfam family.

*Arguments*

- *rfamFamily*: string with the Rfam family accession or ID for which a descriptive summary should be retrieved.

  *Example of usage*
  ```
  ## Obtain a summary for the Rfam family with accession RF00005,
  ## of which several instances were identified in the mitochondrial
  ## DNA of Ashbya gossypii. It corresponds to tRNA.
  rfamFamilySummary(rfamFamily = "RF00005")
  ```

**rfamConsensusSecondaryStructure.** *Purpose*. Retrieves the consensus secondary structure and sequence of the specified Rfam family.

*Arguments*

- *rfamFamily*: string with the Rfam family accession or ID for which a descriptive summary should be retrieved.

- *filename*: optional string specifying the name of a file. If provided, the consensus secondary structure and sequence will be saved to the specified file.

- *format*: string indicating the notation to be used for the RNA secondary structure. It can be either "DB" (extended Dot-Bracket notation; default) or "WUSS" (Washington University Secondary Structure notation).

  *Example of usage*
  ```
  ## Obtain the consensus secondary structure and sequence
  ```

```
## for the Rfam family with accession RF00005 (tRNA)
## in the extended Dot-Bracket format
rfamConsensusSecondaryStructure(rfamFamily = "RF00005", format =
"DB")
```

**rfamSecondaryStructurePlot.** *Purpose*. Plots a diagram of the specified type of the secondary structure of an Rfam family.

*Arguments*

- *rfamFamily*: string with the Rfam family accession or ID for which the secondary structure should be plotted.

- *filename*: optional string specifying the name of a file. If provided, the plot will be saved to the specified file.

- *plotType*: string indicating the desired type of secondary structure diagram. Possible values are "norm" (normal), "cons" (sequence conservation), "fcbp" (basepair conservation), "cov" (covariation), "ent" (relative entropy), "maxcm" (maximum covariance model parse), "rscape" (R-scape analysis of the seed alignment) and "rscape-cyk" (secondary structure predicted by R-scape).

  *Example of usage*
```
## Generate a diagram of the secondary structure of
## the Rfam family with accession RF00005 (tRNA), colored
## by basepair conservation
rfamSecondaryStructurePlot(rfamFamily = "RF00005", plotType = "fcbp")
```
**rfamSecondaryStructureXMLSVG.** *Purpose*. Obtain an SVG file (in XML format) with a representation of the secondary structure of the specified Rfam family.

*Arguments*

- *rfamFamily*: string with the Rfam family accession or ID for which an SVG file the secondary structure should be plotted.

- *filename*: string specifying the path to which the SVG file should be saved.

- *plotType*: string indicating the desired type of secondary structure diagram. Possible values are "norm" (normal), "cons" (sequence conservation), "fcbp" (basepair conservation), "cov" (covariation), "ent" (relative entropy), "maxcm" (maximum covariance model parse), "rscape" (R-scape analysis of the seed alignment) and "rscape-cyk" (secondary structure predicted by R-scape).

  *Example of usage*
```
## Save an SVG file with a diagram of the secondary structure
## of the Rfam family with accession RF00005 (tRNA), colored
## by sequence conservation
rfamSecondaryStructureXMLSVG(rfamFamily = "RF00005",
filename = "test.svg",
plotType = "cons")
```
**rfamSeedAlignment.** *Purpose*. Retrieves the seed multiple alignment of the specified Rfam family. The seed alignment is used to determine the covariance model defining each Rfam family, and comprises only a subset of all the members of each family.

*Arguments*

- *rfamFamily*: string with the Rfam family accession or ID whose seed alignment should be retrieved.

- *filename*: optional string specifying a file to which the seed alignment will be saved if provided.

- *format*: string indicating the desired format for the seed alignment. Possible values are "stockholm" (standard Stockholm format), "pfam" (Stockholm format with alternative secondary structure notation), "fasta" (gapped FASTA format) and "fastau" (ungapped FASTA format).

   *Example of usage*
```
## Obtain the seed alignment of the Rfam family with
## accession RF00005 (tRNA) in the Stockholm format and
## save it to a file
rfamSeedAlignment(rfamFamily = "RF00005", filename = "test.stk", for-
mat = "stockholm")
```

   **rfamSeedTree.** *Purpose*. Retrieves the phylogenetic tree of the seed multiple alignment associated to the specified Rfam family. The tree is retrieved in the NHX format (New Hampshire extended) and saved to a file.
   *Arguments*

- *rfamFamily*: string with the Rfam family accession or ID for which the phylogenetic tree of the seed alignment should be retrieved.

- *filename*: string specifying a file to which the phylogenetic tree will be saved.

   *Example of usage*
```
## Obtain the phylogenetic tree of seed alignment of the
## Rfam family with accession RF00005 (tRNA) and save it
## to a file
rfamSeedTree(rfamFamily = "RF00005", filename = "test.nhx")
```

   **rfamSeedTreeImage.** *Purpose*. Plots the phylogenetic tree of the seed multiple alignment associated to the specified Rfam family.
   *Arguments*

- *rfamFamily*: string with the Rfam family accession or ID for which the phylogenetic tree of the seed alignment should be plotted.

- *filename*: optional string specifying a file to which the plot of the phylogenetic tree will be saved if provided.

- *label*: string indicating the type of labels that should be added to the plot of the phylogenetic tree. Can be either "species" (for labeling with species names) or "acc" (for labeling with sequence accessions).

   *Example of usage*
```
## Plot the phylogenetic tree of seed alignment of the
## Rfam family with accession RF00005 (tRNA) labelled with
## species names
rfamSeedTreeImage(rfamFamily = "RF00005", label = "species")
```

   **rfamCovarianceModel.** *Purpose*. Retrieves the covariance model of the specified Rfam family (generated with the Infernal software).
   *Arguments*

- *rfamFamily*: string with the Rfam family accession or ID for which the covariance model should be retrieved.

- *filename*: string specifying a file to which the covariance model will be saved.

*Example of usage.*
```
## Retrieve the covariance model of the Rfam family with
## accession RF00005 (tRNA) and save it to a file
rfamCovarianceModel(rfamFamily = "RF00005", filename = "test.cm")
```

**rfamSequenceRegions.** *Purpose.* Retrieves all sequence regions encoding an RNA assigned to be a member of the specified Rfam family.

*Arguments*

- *rfamFamily*: string with the Rfam family accession or ID for which the member sequence regions should be retrieved.

- *filename*: optional string specifying the name of a file. If provided, the sequence regions will be saved to the specified file in tab-delimited format.

*Example of usage*
```
## Retrieve the sequence regions belonging to the Rfam
## family with accession RF00177 (small subunit rRNA)
rfamSequenceRegions(rfamFamily = "RF00177")
```

**rfamPDBMapping.** *Purpose.* Retrieves entries of the PDB database with the experimentally solved 3D structure of members of the specified Rfam family, with correspondences between residues of the PDB structure and positions in the covariance model of the Rfam family.

*Arguments*

- *rfamFamily*: string with the Rfam family accession or ID for which the matching PDB entries should be retrieved.

- *filename*: optional string specifying the name of a file. If provided, the matching PDB entries will be saved to the specified file in tab-delimited format.

*Example of usage*
```
## Retrieve the PDB entries with structures of members
## of the Rfam family with accession RF00005 (tRNA)
rfamPDBMapping(rfamFamily = "RF00005")
```

## Case study: Analysis of the SARS-CoV-2 genome

In order to demonstrate the functionalities of our package, we used rfaRm to analyze the reference genome of SARS-CoV-2 (RefSeq accession number NC_045512.2). For this, we first identified the non-coding RNA elements present in the SARS-CoV-2 genome. We then extracted and plotted information concerning the RNA families found to be present in the genome and showed how such information can be further processed with other Bioconductor packages.

Even though the genome is 29,903 bases long, such a sequence can be directly processed by rfaRm thanks to the internal splitting into fragments of 10,000 bases. We chose an overlap between consecutive fragments of 3,000 nucleotides to minimize the risk of missing hits present at the boundaries between fragments. We performed the search with and without clan competition to compare the results. An illustration of these queries implemented using R is provided hereafter:
```
library(rfaRm)
## Read genome from FASTA file
library(seqinr)
sars_cov_2_genome <- unlist(read.fasta("sars_cov_2_genoma.fasta",
seqtype = "DNA",
as.string = TRUE))
```

```
## Convert DNA string to RNA string
sars_cov_2_RNA_genome <- gsub("t", "u", sars_cov_2_genome, ignore.
case = TRUE)
## Search for Rfam families hits in the whole genome without clan
competition
sars_cov_2_Rfam_hits <- rfamSequenceSearch(sars_cov_2_RNA_genome,
fragmentsOverlap = 3000,
clanCompetitionFilter = FALSE)
length(sars_cov_2_Rfam_hits)
## Search for Rfam families hits in the whole genome with clan
competition
sars_cov_2_Rfam_hits_2 <- rfamSequenceSearch(sars_cov_2_RNA_genome,
fragmentsOverlap = 3000, clanCompetitionFilter = TRUE)
length(sars_cov_2_Rfam_hits_2)
## Search for Rfam families hits in the whole genome without clan
competition
sars_cov_2_Rfam_hits_2 <- rfamSequenceSearch(sars_cov_2_RNA_genome,
fragmentsOverlap = 3000, clanCompetitionFilter = FALSE)
length(sars_cov_2_Rfam_hits_2)
```

The search without clan competition returned a total of 7 RNA families in the SARS-CoV-2 genome (Table 1).

After extracting the IDs of all families, we performed queries for all of them to obtain a brief description with relevant information about the type and role of each RNA family:

```
## Extract Rfam IDs
rfamIDs_noClanCompetition <- as.character(lapply(sars_cov_2_Rfam_-
hits, "[[", 2))
rfamIDs_ClanCompetition <- as.character(lapply(sars_cov_2_Rfam_-
hits_2, "[[", 2))
## Iterate over set of Rfam IDs and retrieve a summary for each one
summary_list <- list()
for (id in rfamIDs_noClanCompetition) {
summary <- list(rfamFamilySummary(id))
names(summary) <- id
summary_list <- c(summary_list, summary)
}
```

All identified families were non-coding RNA elements typically found in the genome of beta-coronaviruses:

1. bCoV-5UTR: 5' untranslated region comprising 150–200 nucleotides found in beta-coronaviruses.

2. Sarbecovirus-5UTR: 5' untranslated region specific of SARS beta-coronaviruses.

**Table 1. Non-coding RNA elements present in the SARS-CoV-2 genome.**

| RNA family | Start position | End Position |
|---|---|---|
| bCoV-5UTR | 2 | 300 |
| Sarbecovirus-5UTR | 2 | 300 |
| Corona_FSE | 13470 | 13551 |
| bCoV-3UTR | 29519 | 29871 |
| Sarbecovirus-3UTR | 29537 | 29871 |
| Corona_pk3 | 29604 | 29663 |
| s2m | 29728 | 29770 |

For each RNA element, the Rfam family ID and start and end positions in the SARS-CoV-2 genome are indicated.

3. Corona_FSE: stem loop conserved amongst coronaviruses that can promote ribosomal frameshifting.

4. bCoV-3UTR: 3' untranslated region comprising 300–500 nucleotides found in beta-coronaviruses.

5. Sarbecovirus-3UTR: 3' untranslated region specific of SARS beta-coronaviruses.

6. Corona_pk3: conserved pseudoknot of approximately 55 nucleotides found in the 3' untranslated region of coronaviruses.

7. s2m: motif of unknown function found in the 3' untranslated region of astroviruses, coronaviruses and equine rhinoviruses.

On the other hand, the search with clan competition enabled only returned 3 hits, corresponding to Sarbecovirus-5UTR, Corona_FSE and Sarbecovirus-3UTR. bCoV-5UTR and bCoV-3UTR are discarded because they are essentially the same hit as the Sarbecovirus-5UTR and Sarbecovirus-3UTR respectively, with the only difference that the latter are versions found specifically in SARS beta-coronaviruses. The Corona_pk3 and s2m motifs were also omitted, since they are comprised within the larger Sarbecovirus-3UTR.

Knowing the IDs of the RNA families of interest, more detailed information can be easily retrieved through a set of query functions. As an example, we acquired detailed information about the Sarbecovirus-5UTR (with Rfam accession number RF03120).

First, we extracted the consensus sequence and secondary structure for the family:

```
## Retrieve consensus sequence and secondary structure and save them
to a file in the
## extended dot-bracket format
rfamConsensusSecondaryStructure("RF03120", filename = "RF03120_cons.
txt", format = "DB")
## [1] "AuauuAgGcuuuuACCuaccCaGGaa..aagCcAAccAA.uuUcGauCuCUUGUaGauCU-
GuuCUcUAAAcGa.
aCUUUAAAA......UCuGcGuggCuGUCgCucgGCUGcAUGCcuaGcGCacccaCgCaGUAUAAauAa-
UAAuaAAuUUUAcUGuCGuuGaCagGgaaCgaGUAAcUcGuCcauCuuCuGCAGgCuGCUcaCG-
GUUUCGUCCGugUUGCaGcCGAUCAUCaGCacacCcAGGUUUcGUCCgGguguGaCCGAAAGGuaa-
GaUgGaGaGCCucGucCcuGGuuuCaaCGaGAAAA"
## [2] "......<<<<<<<.<<<. . . .>>>>>..>>>>>. . . . . .. . ..<<<<<. . .
..>>>>>.<<<<. . .. . ..>>.>>. . . . . .. . .. . .<<<<<<<<.<<.<<<<.
<<<. . . ..>>>.>>>>>.>>>>>>>. . . . .. . .. . ...  . .. . . ..
((((((((((((.(((((. . .(((.(((.((((<<<..<<<<<<.<<<<<. . . . .>>>>>..
>>>>>>. . .. . .>>><<<<<<.<<. . .. . .>>>>>>>>>><<<. . . .
>>>)))).)))))).)))))))))))...)))))))... .."
```

Next, we extracted the seed multiple sequence alignment of the family:

```
## Retrieve seed multiple sequence alignment and save it to a FASTA
file
rfamSeedAlignment("RF03120", filename = "RF03120_seedAlgn.fasta",
format = "fasta")
```

The information from the consensus secondary structure and the multiple seed alignment can be combined and visualized with the R4RNA package (Fig 2):

```
## Read consensus secondary structure and multiple alignment
library(R4RNA)
library(Biostrings)
secondaryStructureTable <- readVienna("RF03120_cons.txt")
seedAlignment <- readBStringSet("RF03120_seedAlgn.fasta")
## Make a helix plot of the consensus secondary structure and annotate
it with information
## from the seed alignment
```

```
plotCovariance(seedAlignment, secondaryStructureTable, grid = TRUE,
line = TRUE, arrow = TRUE,
legend = FALSE, cex = 3)
```

It is also possible to generate plots of the secondary structure annotated with different types of information by specifying the "format" argument. Possible values are:

- norm: default type with no annotation

- cons: sequence conservation

- fcbp: basepair conservation

- cov: covariation

- ent: relative entropy

- maxcm: maximum covariance model parse

- rscape: R-scape analysis of the seed alignment of the family

- rscape-cyk: secondary structure predicted by R-scape from the seed alignment of the family

For example, in order to generate a plot of the secondary structure of the Sarbecovirus-5UTR annotated with sequence conservation (Fig 1B), the following example code can be used:

```
## Generate a plot of the secondary structure of Rfam family RF03120
annotated with
## sequence conservation and save it to a PNG file
rfamSecondaryStructurePlot("RF03120", filename = "RF03120_cons.png",
plotType = "cons")
## The plot can also be saved to an SVG file, which is useful to save
images into an
## editable vector-based format
rfamSecondaryStructureXMLSVG("RF03120", filename = "RF03120_cons.
svg", plotType = "cons")
```

A plot of the phylogenetic tree of the seed alignment can be easily generated and labeled with species names or sequence accession numbers (Fig 1C):

```
## Generate a plot of the phylogenetic tree of Rfam family RF03120
labeled with species
## names and save it to a GIF file
rfamSeedTreeImage("RF03120", filename = "RF03120_tree.gif", label =
"species")
```

Additionally, the phylogenetic tree can be retrieved in the New Hampshire Extended (NHX) format. The tree can then be read and processed with other software, such as the treeio R package:

```
## Save the phylogenetic tree of Rfam family RF03120 to a file in the
NHX format
rfamSeedTree("RF03120", filename = "RF03120_treeNHX.nhx")
## Read the tree as a treedata object
library(treeio)
treeioTree <- read.nhx("RF03120_treeNHX.nhx")
## Print a summary of the tree
as.phylo(treeioTree)
##
## Phylogenetic tree with 19 tips and 17 internal nodes.
##
## Tip labels:
```

```
## _DQ022305.2/1-295_Bat_SARS_coronavirus_HK...1, _DQ648857.1/1-
297_Bat_CoV_279/2005.1, _MG772934.1/1-298_Bat_SARS-like_coronavirus_
{}.1, _MT345841.1/1-293_Severe_acute_respiratory_syndrome_corona-
virus_2.6, _MT344963.1/1-299_Severe_acute_respiratory_syndrome_coro-
navirus_2.2, _MT345869.1/1-
293_Severe_acute_respiratory_syndrome_coronavirus_2.5,...
## Node labels:
##, 0.780, 0.940, 0.800, 0.890, 1.000,...
##
## Unrooted; includes branch lengths.
```

## Conclusion

The rfaRm R package provides an easy-to-use client-side interface to the Rfam database, enabling users to access it programmatically and therefore bypassing the limitations of interactive access through the web interface or the requirement to install and search the database locally. Programmatic access allows the identification of non-coding RNA across entire genomes. The package is designed to interoperate with existing software by returning data into formats directly readable by other tools and R packages and is available as part of the Bioconductor project, which facilitates its integration within workflows and pipelines for the analysis of genomic data. We believe the package will provide a useful resource for the community of RNA Bioinformatics, whose interest in the tool has been demonstrated by the considerable number of downloads of the package in spite of its still short lifetime.

## Author Contributions

**Conceptualization:** Lara Sellés Vidal, Rafael Ayala.

**Formal analysis:** Lara Sellés Vidal, Rafael Ayala.

**Funding acquisition:** Guy-Bart Stan, Rodrigo Ledesma-Amaro.

**Investigation:** Lara Sellés Vidal, Rafael Ayala.

**Methodology:** Lara Sellés Vidal, Rafael Ayala.

**Software:** Lara Sellés Vidal, Rafael Ayala.

**Supervision:** Guy-Bart Stan, Rodrigo Ledesma-Amaro.

**Validation:** Lara Sellés Vidal, Rafael Ayala.

**Visualization:** Lara Sellés Vidal, Rafael Ayala.

**Writing – original draft:** Lara Sellés Vidal, Rafael Ayala.

**Writing – review & editing:** Lara Sellés Vidal, Rafael Ayala, Guy-Bart Stan, Rodrigo Ledesma-Amaro.

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
