## [Decision Letter · Decision Letter 0]

21 Dec 2020

PONE-D-20-33168

rfaRm: an R client-side interface to facilitate the analysis of the Rfam database of RNA families

PLOS ONE

Dear Dr. Selles Vidal,

Thank you for submitting your manuscript to PLOS ONE. After careful consideration, we feel that it has merit but does not fully meet PLOS ONE’s publication criteria as it currently stands. Therefore, we invite you to submit a revised version of the manuscript that addresses the points raised during the review process.

The paper is quite interesting. Please revise it as much as possible following the suggestion of the Reviewer. A couple of examples should be presented by using new data or publicly available dataset to demonstrate the power of the software/database.

We look forward to receiving your revised manuscript.

Kind regards,

Zhong-Hua Chen, Ph.D.

Academic Editor

PLOS ONE

Journal Requirements:

Reviewers' comments:

Reviewer's Responses to Questions

**Comments to the Author**

1. Is the manuscript technically sound, and do the data support the conclusions?

Reviewer #1: Partly

2. Has the statistical analysis been performed appropriately and rigorously? 

Reviewer #1: Yes

3. Have the authors made all data underlying the findings in their manuscript fully available?

Reviewer #1: Yes

4. Is the manuscript presented in an intelligible fashion and written in standard English?

Reviewer #1: Yes

5. Review Comments to the Author

Reviewer #1: 1. It is necessary to replace high-resolution figures.

2. in addition to the case study, you had better to add descriptions for specific functions of your package, such as usage and arguments of rfamSequenceSearch, rfamFamilySummary, rfamSeedAlignment.

6. PLOS authors have the option to publish the peer review history of their article (what does this mean?). If published, this will include your full peer review and any attached files.

Reviewer #1: No

---

## [Author Response · Author response to Decision Letter 0]

22 Dec 2020

The editor requested the addition of some more examples. These have been added as individual examples of usage for each function available in the package. More usage examples with real, publicly available data can be seen at the manual and vignette of the package

The reviewer had two comments:

1) replacing of figures by high resolution tif files. This has been done.

2) addition of descriptions of each function of the package (including arguments and usage). This has been done.

---

## [Editor Report · Decision Letter 1]

28 Dec 2020

rfaRm: an R client-side interface to facilitate the analysis of the Rfam database of RNA families

PONE-D-20-33168R1

Dear Dr. Selles Vidal,

We’re pleased to inform you that your manuscript has been judged scientifically suitable for publication and will be formally accepted for publication once it meets all outstanding technical requirements.

Kind regards,

Zhong-Hua Chen, Ph.D.

Academic Editor

PLOS ONE
---

## [Editor Report · Acceptance letter]

29 Dec 2020

PONE-D-20-33168R1 

rfaRm: an R client-side interface to facilitate the analysis of the Rfam database of RNA families 

Dear Dr. Selles Vidal:

I'm pleased to inform you that your manuscript has been deemed suitable for publication in PLOS ONE. Congratulations! Your manuscript is now with our production department. 

Kind regards, 

on behalf of

Dr. Zhong-Hua Chen 

Academic Editor

PLOS ONE